# Conversion of Mixtures of Soybean Curd Residue and Kitchen Waste by Black Soldier Fly Larvae (*Hermetia illucens* L.)

**DOI:** 10.3390/insects13010023

**Published:** 2021-12-24

**Authors:** Xinfu Li, Zhihao Zhou, Jing Zhang, Shen Zhou, Qiang Xiong

**Affiliations:** 1College of Food Science and Light Industry, Nanjing Tech University, No. 30 Puzhu Road South, Nanjing 211800, China; lixinfu316@126.com (X.L.); zhou_shen0629@126.com (S.Z.); zhouzhihao91@126.com (Z.Z.); 2College of Biotechnology and Pharmaceutical Engineering, Nanjing Tech University, No. 30 Puzhu Road South, Nanjing 211800, China; zhangjing3737@126.com

**Keywords:** black soldier fly, conversion, soybean curd residue, kitchen waste, bioconversion (biomass production)

## Abstract

**Simple Summary:**

The black soldier fly (BSF) is a viable solution for food waste management and can provide a sustainable protein source to feed the growing global population. However, the growth performance of BSF larvae (BSFL) is greatly influenced by the rearing substrate. An imbalanced diet caused by the utilization of single substrate could be solved using a mix of different waste types and formulating a more balanced diet, which would provide a more reasonable nutritious and balanced energy feed for larval growth. This study focused on the effects of different proportions of mixtures of soybean curd residue (SCR) and kitchen waste (KW) on the performance of BSFL. The key findings of this study are: the highest larval biomass (30.32 g fresh and 11.38 g dry mass), bioconversion rate (18.54%) and larval crude lipid (45.91%), and the lowest feed conversion ratio (FCR) (2.51) were obtained when BSFL were fed with 30% SCR and 70% KW.

**Abstract:**

The production of insect biomass from organic waste is a major challenge in terms of reducing the environmental impacts of waste and maintaining feed and food security. The feasibility of the co-conversion of soybean curd residue (SCR) and kitchen waste (KW) to breed black soldier fly (BSF, *Hermetia illucens*) larvae was evaluated so as to enhance biomass conversion efficiency and supply animal feed and allow it to be used in biodiesel production. Co-digestion was found to significantly increase larval yield, bioconversion rate, and bioaccumulation of lipid. Partial least squares regression showed that the conversion of 30% SCR with 70% KW is an appropriate proportion. The appropriate performance parameters of BSF were: survival rate (98.75%), prepupal rate (88.61%), larval biomass (30.32 g fresh and 11.38 g dry mass), bioconversion rate (18.45%), efficiency conversion of ingested food (ECI) (28.30%), and FCR (2.51). Our results show that conversion of mixtures (e.g., SCR with KW) by BSF larvae (BSFL) could play an important role in various organic materials management.

## 1. Introduction

Soybean curd residue (SCR), also known as okara, is the main by-product from processed soy products such as tofu, soy milk, soy sauce, miso, natto, etc., is often treated as waste [1]. Approximately 1.1–1.2 kg of fresh SCR is produced from every kilogram of soybeans processed into tofu or soy milk. In 2012, more than 3,900,000 tons/year SCRs were produced in China, Japan, Korea, and some other regions of the world [2]. Kitchen waste (KW) is a mixture of various substances mainly including cooked wheaten food, vegetables, rice, fish, meat, oil, fruits and animal by-products, and its main sources are restaurants and kitchens [3]. KW is not only a carbon source, but it is also an excellent nutrient source [4]. Approximately 1.3 billion tons of KW are produced in the food supply chain every year, and this amount increases as the economy and population increase [5]. Current management operations for SCR and KW apply it in the breeding industry (e.g., livestock and poultry breeding), on arable land as organic fertilizers, as landfill, or as direct combustion, which pollutes the environment [1,3,6], resulting in public nuisance and environmental problems [7]. Accordingly, there is an urgent need for new technologies for the rational treatment of organic waste recycling, in order to alleviate its adverse impact on the natural environment.

Insects have gained much attention as a valuable protein-rich and fat-rich biomass source from organic waste [8,9]. *Hermetia illucens* (Diptera: Stratiomyidae), known as the black soldier fly (BSF), have received widespread attention over the last decade, and their early introduction for waste treatment can be traced back to the 1990s [10]. BSF larvae (BSFL) can feed on a number of different substrates, such as decomposing fruits and vegetables, animal feces, human excreta, municipal organic waste, distillery grains, and even crop straw [11,12]. To promote sustainable management of organic materials and increase the added value of processing waste, BSFL bioconversion is regarded as a promising technique to convert varietal organic waste into alternative protein-rich and fat-rich raw materials [2,13,14].

Nonetheless, the growth performance of BSFL in accordance with the accumulated yield (e.g., larval mass and bioconversion) is affected by the larval rearing materials [15]. SCRs provide a rich source of protein, fiber, fat/oil, and carbohydrates [1], making them a conceivable feedstock for BSFL [16]. The influence of fiber food by-products on larval growth was examined, and the results showed that larvae harvest mass and yield increased with decreasing carbon-to-nitrogen ratio [17]. Dietary fiber limits the development of larval growth and the development of *Tenebrio molitor* (Coleoptera: *Tenebrionidae*), which is another species used to obtain protein source from waste [18]. Therefore, finding an effective method of digestion for SCR is the premise of using it as BSFL feed substrate. The pH seems to play an important role on the BSFL and the larvae can modulate it to maintain optimal conditions [19,20].

Previous studies on synergistic effects, during BSFL biomass production with the co-conversion of organic waste, have been fruitful (Table 1). Accordingly, the co-digestion of fiber-rich or carbohydrate-rich portions with a stable formation of larval biomass nutritional composition from BSFL has been improved. Rehman et al. [21] observed that the co-digestion method appreciably improved the BSFL process performance (e.g., the larval production, waste reduction, and feed conversion ratio (FCR)). Batch experiments with various mixtures of dairy manure (fiber-rich) and chicken manure showed that 4:6 was the optimal proportion. Rice bran (high-fiber-rich) as co-digestion substrate was added to the BSFL rearing substrates (chicken and pig manure), the addition ratio of about 15% exhibited the best transformation efficiency [22]. The co-digestion experiment regarding rice straw and restaurant food waste showed that the conversion rate was the highest when the ratio of rice straw and food waste was 3:7 [23]. Co-substrate of fecal sludge and organic waste (30%) can be used as a recommendation for large-scale production of BSFL [24]. The impact of co-digestion of dairy manure with SCR showed that the mixture of dairy manure and SCR in a proportion of 2:3 was confirmed to be the favorable outcome when using such measures [7].

Therefore, investigators fed BSFL with SCR mixed organics to improve the utilization efficiency of the two substrate nutrients [7]. Mixing SCR with low fiber matrix can increase the nutrient content in larval matrix feed, and the final effect on transformation efficiency has not been considered. KW has low sourness with less fiber, a high protein content, and rich organic matter [3]. However, to the best of our knowledge, co-conversion between SCR and KW of BSFL biomass remains undocumented. Therefore, enhancing the co-conversion of SCR (rich fiber) with KW for the cultivation of BSFL was investigated.

In this study, the effects of different proportions of mixtures of SCR and KW on the performance of BSFL were determined. The prime objective was to elucidate the correlation between the different substrates of organic matter on larval growth performance (biomass conversion ratio, larval mass), waste reduction efficiency, and nutritional composition and find a high-value technology for co-digestion of organic waste by BSF.

## 2. Materials and Methods

### 2.1. Raw Materials

Eggs of the BSF (*H. illucens*) were obtained from Taizhou Younong Biotechnology Co., Ltd. (Tanzhou, China). SCR was provided by the Zhuquan farmers’ markets in Nanjing, China. KW was sourced from the food provision outside homes as well as household sources, separated in cities. In this study, KW was also supplied by Taizhou Younong Biotechnology Co., Ltd., and was collected from restaurants in Taizhou city, China. The relative contents of the main components of SCR and KW were measured before use, which are listed in Table 2.

### 2.2. Conversion of SCR and KW

Batch tests on SCR and KW materials were conducted to evaluate the effects of mixing ratio on the yield of BSFL biomass accumulation. Fresh SCR and KW were fed BSFL as feed stock and mixed in a form at the appropriate mass ratio (wt/wt). Six mixing ratios of SCR:KW were assessed: (0:100) M0, (20:80) M20, (30:70) M30, (40:60) M40, (50:50) M50, (100:0) M100.

In the present study, the 6-day old larvae were fed on standard colony diet before use [29]. Larvae and different substrates were laced in each 3 L plastic container. Based on preliminary analysis, approximately 200 of the 6-day old BSFL (sixth group, total 1200) were inoculated into each vessel with the recording of a continuous date and time. These studies were carried out in a greenhouse at 28 ± 2 °C with 70% moisture conditions. Batch tests were conducted to determine the larval growth and development. The surveys were performed in triplicate for each pure feed and mixture feed.

Larval transformation was interrupted when feeding after substrate was added throughout the 12 days. The larvae were manually picked from the residues and water washed. Then, all harvested larvae were inactivated at 110 °C for 10 min and dried at 50 °C to a constant mass. The remaining residue was air dried at 105 °C to constant final mass. The development time, fresh larval mass, survival rate, and fresh residue mass were measured after the transformation with BSFL. The BSFL process performance such as larval growth (biomass conversion ratio, final larval mass), waste reduction and larval composition were detected after the transformation. A suitable mixing ratio of SCR and KW substrates for BSFL co-digestion was investigated.

### 2.3. Chemical Analysis

The total water content and dry mass were measured by drying (105 °C) under atmospheric pressure according to Chinese National Standard GB 5009.3-2010. The ash content was determined following National Standard GB 5009.4-2010. Crude protein (CP) was measured using the Kjeldahl method and a conversion factor of 6.25 was calculated by using the method in GB/T 5009.5-2010. Crude fat (CF) of larvae and feedstock was determined using Soxhlet extraction to GB/T 5009.6-2003. Total phosphorus (TP), total nitrogen (TN), and total organic carbon (TOC) were measured in accordance with the guidelines of the Chinese Agricultural Standards (NY 525-2011). Carbon-to-nitrogen (C/N) ratios were calculated based on the percentages of carbon and nitrogen by mass in sucrose and urea, on a dry matter basis. The carbon-to-nitrogen ratio (C/N) was calculated by dividing the percentage of organic carbon by the percentage of total nitrogen, on a dry matter basis. The percentage of organic carbon was estimated by a fixed factor that typically ranges from 1.4 to 1.8 [27].

### 2.4. Processing Parameters

The BSFL biomass production parameters included survival rate, prepupal rate, fresh larval mass, dry larval mass, dry mass reduction, bioconversion rate, FCR, and efficiency of conversion of ingested food (ECI) based on previous research [2,21].
Survival rate (%) = [number of larvae at the end of the test/number of larvae at the beginning of test] × 100%(1)
Prepupal rate (%) = [the number of prepupa at the end of the test/the number of larvae at the end of the test] × 100%(2)
Dry mass reduction (%) = [(mass of feed at the beginning of the test (g) − mass of residue at the terminations of the test (g))/mass of feed at the beginning of the test (g)] × 100%(3)
Bioconversion rate = [total larval biomass (g)/feed added (g)] × 100%(4)
FCR (g/g) = mass of ingested feed (g)/mass gained (g)(5)
ECI (g/g) = mass of BSFL/mass of ingested feed × 100%(6)

### 2.5. Calculation and Statistical Analysis

The statistical analysis was performed using SPSS 16.0 (SPSS Inc., Chicago, IL, USA). Analysis of results of all experiments was undertaken by one-way analysis of variance (ANOVA). Post-hoc multiple comparisons were determined by use of Tukey’s test with the level of significance set at *p* < 0.05. Nutritional constituents of mixed organics. Partial least squares regression (PLSR) analyses were developed by using Unscrambler software 9.7 (AS, Oslo, Norway) as described previously [30]. The mean values of the physico-chemical indices and process performance were processed using ANOVA-PLSR.

## 3. Results

### 3.1. Raw Material Properties

Relative amounts of the representative main components of SCR and KW and mixtures in different ratios were measured before use (Table 2). SCR mainly included crude protein (23.24%), crude fiber (22.18%), total carbohydrates (30.15%), and crude fat (9.19%). KW includes the perishable food waste produced by typical families and mainly includes uneaten portions of meals, vegetables, fruit peels, leftovers and waste food, crude protein (25.41%), crude fiber (12.34%), total carbohydrates (28.92%), and crude fat (13.37%).

### 3.2. Survival Rate, Prepupal Rate, and Larval Production

All the ratios of SCR and KW were adopted by the BSFL for their development; however, the survival rate, prepupal rate, and larval dry mass were affected by adding different amounts of KW to the SCR (Table 3). The survival rate was significantly lower in the M0 and M20 than in the other groups, but it improved when the content of SCR reached 30% or above. However, there was no significant difference between M30, M40, M50 and M100. There was a fluctuation in the survival rate from 86.00% in M20 substrate to 98.75% in M30. The lowest prepupal rate was 80.36% in M100, but when fed with mixtures of SCR and KW, the prepupal rate increased significantly. The highest prepupal rate was 90.42% in M20. Regarding larval production in the fresh and dry BSFL yield in the co-conversion experiments with mixtures, there are two significant results shown in Table 3: initially, the value decreased from M0 (19.83 and 7.56, respectively) to M30.

### 3.3. Dry Mass Reduction, Bioconversion Rate, and FCR

The results of the waste mass reduction, bioconversion, conversion ratio (FCR), and conversion efficiency of ingested (ECI) feed were measured using dry mass (DW) bases (Table 3). Dry mass reduction and FCR were significantly increased with the addition of SCR, while M100 showed the highest percentage increases (58.36% and 3.44, respectively). Conversely, the highest ECI value was in M30 (28.30%), which then decreased (*p* < 0.05) with increasing proportions of SCR, reaching its lowest value (19.07%) in the M100 group.

Referring to Table 3, the fluctuation of bioconversion rate occurred in two stages: in the first stage, ranging from M0 to M30, the bioconversion rate was increased from 14.31% to 18.54%; in the second (from M40 to M100), the bioconversion rate was decreased from 18.45% to 15.14% (Table 3). It was particularly evident that the highest percent bioconversion rate among the co-conversion mixtures was shown in M30 (18.54%), with barely noticeable deviations compared with M40 (18.45%). The biological conversion rate of larvae was higher than the conversion rate of SCR and dairy manure (14.60%) [7].

### 3.4. Crude Protein and Crude Fat Content of BSFL

The protein and lipid yields from BSFL reared using SCR, KW and various mixed substrate are illustrated in Figure 1. The average CP content of the BSFL was 43.99%. The differences in protein content in BSFL reared at different ratios on different substrates were small (Figure 2 and Appendix A). The initial protein values of the M0 and M20 were 44.17%, and 44.65%, respectively. There was a fluctuation in the protein content from 44.50% in M30 to 43.91% in M100, and the protein content of BSFL reared with pure SCR was the lowest. Furthermore, the crude lipid of the co-conversion mixtures was significant change (*p* < 0.05) than those in the pure SCR (M100, 30.51%) and pure KW (M0, 32.71%). There was no statistically significant difference (*p* > 0.05) between the highest crude lipid content (M30, 35.00%) and the M40 (34.76%).

### 3.5. Relationship between Organic Matter Nutrients and Growth Performance of BSFL

The possible relationship between the characteristics of the nutritional constituents of mixed organics and growth performance of BSFL was revealed. ANOVA-PLSR was generated to process the mean data accumulated from the organic matter nutrients and growth performance of BSFL. The *X*-matrix constituted the measurements of nutritional constituents of mixed organics. The *Y*-matrix constituted the design variables based on growth performance (e.g., survival rate, prepupal rate, larval biomass, and bioconversion rate). In this analysis, the resultant correlation loading plots of PC1 and PC2 describe 78% and 88% of cross-validated variance, respectively. The inner ellipses in the plot indicated 50% of the explained variance and the outer ellipses can explain 100% of the variance. Most of the loading plots are situated among the small and the large circles, which indicate that the data are well illustrated by the PLSR model [31].

As shown in Figure 2, the mixed organics on M0 on the negative region of PC1 have good correspondence with the crude protein, crude fat, crude ash, and ECI. M100 is located in the positive region of PC1 surrounded by FCR, larval crude protein, dry mass reduction, crude fiber, and total carbohydrates. M30, M40, and M50 are located in the positive region of PC1 and PC2, and they are closely related to the replies of larval crude lipid, fresh larval mass, bioconversion rate, and prepupal rate. According to Figure 2, M30 is significantly correlated with most of the growth performance indices.

## 4. Discussion

The BSFL can be used to efficiently convert organic waste into insect biomass that is rich in protein and fat for feed and non-food application. Therefore, the present study was conducted to evaluate the co-digestion of BSFL reared on SCR and the mixtures with KW in different proportions and its effects on growth performance.

### 4.1. Process Performance

Previous research explored the influences of different substrates on the growth performance of BSFL [32,33,34], insect diet being the most important factor for influencing the growth of insect-based products [35,36]. A significantly higher prepupal yield (*p* < 0.05) was detected on blending fecal sludge with other waste feedstock as a co-digestion [24]. In the present study, the effects of dietary composition of BSFL on feed conversion efficiency and growth performance were assessed.

From M0 to M40, the mass continuously increased with the higher ratio of SCR nutritional constituent, which was the highest level when the SCR reached 30–40% in M30 and M40 mixed organics (Table 3). As illustrated in Figure 2, the locations of fresh larval mass and dry larval mass were similar to mixtures M30, M40, and M50 (and particularly M30), confirming that these variables exhibited good correspondence. The excessive SCR impaired the normal development of insect growth of BSFL. Feed quality affects the growth performance of insects, and co-conversion provides more balanced nutrition. Therefore, the transformation efficiency of larvae is effectively improved [37]. 

The survival rate ranged from 81.50 to 99.50% (Table 3), particularly in M0 and M20, it was lower than the 94.00–98.00% reported for SCR and artificial feed [2], 91.20–99.30% for dairy manure and SCR [7]. However, this was higher than the results of a previous investigation in which the survival rate of BSFL ranged from 71.25% to 84.50% on dairy manure [7], and 82.20 to 87.80% on the manure of chick and cow [37]. These differences in survival rate can be intended considering that process elements are influenced by many factors, including feed stock composition, diet and temperature, and relative humidity [7,38]. Thus, the nutrient composition and the mixed ratio of rearing mixture of feeding substrates have a great impact on the production characteristics of BSFL and separated larvae body mass [33]. Cammack and Tomberlin [39] found that the larvae reared on a balanced diet grew the fastest on the lowest amount of food and had the highest survival rate. In this study, SCR mixed in a ratio exceeding 30% (M30) resulted in a better survival rate of 97.75–99.50%, indicating that the optimum nutritional constituent balance in mixed organics.

Nutritional analysis evaluated that the different ratios of mixtures affected the chemical composition of BSFL. Larval crude lipid exhibited a significantly positive influence on M30. When increasing the SCR ratio from 20% to 30%, the insect fat relative content increased from 34.76% to 35.39%. The fat compositions of larvae mainly depend on diet, with values ranging from 31.70% to 47.60% crude protein and 11.8% to 34.3% crude fat in previous research [2,36,40,41]. The crude fat contents of the mixtures increased from 42.50% (M0) to 45.91% (M30), which was higher than the average concentrations of BSFL reared on three by-products [42]. The percentage of crude fat of BSFL was similarly influenced by dietary treatment, the co-digestion of SCR and KW was beneficial to the accumulation of fat in larvae, a significant increase in crude lipid output could be calculated but hindered the accumulation of protein. Lopes et al. [26] found that adding small amounts (<15%) of protein-rich substrate (aquaculture waste) was demonstrated to be very beneficial for process performance in BSFL composting. Major differences in larval rearing conditions and methods used to perform nutritional analyses potentially influenced the reported protein composition (12.9–78.8%) of the BSFL [43]. The fat content of larvae in this study (31.50–35.00%) was lower than that found in previous studies (31.70–47.60%) [33]. Proteinogenic nitrogen energy was consumed during insect transformation. Therefore, the high protein substance (23.24–25.41%) in this study may not be conducive to larval development and fat accumulation.

Most growth parameters, such as prepupal rate, bioconversion rate, and mass reduction are shown to make a significant contribution to BSFL. The prepupal rate of BASL fed on pure substrate (SCR 80.36% and KW 83.48%, respectively) was significantly lower than that in the mixed substrate group, which may be related to the advantage bestowed by the mixture. Spranghers et al. [36] fed BSFL with restaurant waste and observed the slowest emergence of prepupal larvae. This could be due to the high amount of grease in the substrate, which is difficult to process for BSFL [44]. The prepupal rate 83.48% in M0 was lower, which may be related to the relatively hard and adhesive structure of pure KW. BSFL cannot find enough space to grow within this structure. These results suggest that the performance of BSFL treatment facilities was increased by designing biowaste mixtures based on fiber content.

The bioconversion on the dry mass-based mixture was 14.31% in pure KW M0 and 15.14% in SCR M100, but a significantly higher bioconversion of 18.54% was found in the mixture of M30 (Table 3). The co-conversion mixtures had beneficial effects on the bioconversion and FCR in the current investigation. Rehman et al. [7] found that bioconversion in BSFL composting was higher on SCR than on dairy manure, while mixtures of the two substrates yielded even greater bioconversion. The addition of rice bran could promote the transformation of chicken manure and pig manure by BSFL, and at a proportion 15% the conversion rates were the highest (19.74% and 19.25%, respectively) [22]. The value was higher than that found in the study of Lalander and coworkers [27], wherein the biological conversion rate was 13.9% in BSFL composted with food waste, 12% in municipal organic waste [45], and 3.9–6.9% in SCR assisted with *L. buchneri* and artificial feed [2]. The FCR value indicates that larval development with mixtures is more efficient at converting feed into biomass compared to larval feeding on pure substrates with BSFL (Table 3). The FCR was stated previously in chicken manure (5.6) and dairy manure (10.3) on the dry mass base [21], and 8.0–9.8 in co-digestion mixtures of SCR assisted by *L. buchneri*, and artificial feed [2]. The ECI on pure substrate (SCR 21.90% and KW 19.07%, respectively) was significantly lower in BSFL using mixed treatments (M20, M30, M40, and M50) (Table 3). ECI gives a rough overall measure of the insect’s ability to use ingested food for its growth [46].

In this study, the range of dry matter reduction was 32.71–58.36%. These findings are similar to the former results, which showed that the dairy manure mass reduction was 34.00–58.00% [47], 31.00–61.70% in chicken manure, and 28.00–53.40% in swine manure [48]. BSFL has the ability to reduce dry matter in three different animal manures (swine manure, chicken manure, and dairy manure), but the dry matter reduction of dairy manure was the lowest, as it contained more fiber and fewer nutrients [49]. In the present study, the dry matter reduction increased with the proportion of SCR. In this study, the dry matter reduction significantly increased (*p* < 0.05) with increasing proportion of SCR, and it reached the highest values in the M100 group.

### 4.2. Co-Conversion of Different Treatment Mixtures of BSFL

PLSR was conducted to distinguish the most important parameters affecting BSFL growth performance. M30, M40, and M50 are distributed in the same region, surrounded mainly by the prepupal rate, bioconversion rate, fresh larval mass, dry larval mass, and larval crude fat. These variables have good correspondence with these mixtures, particularly M30 (with the proportion of SCR and KW of 30:70). M30 returned the optimal growth performance; furthermore, in the formulation feeding experiments, crude fiber, total carbohydrates and FCR are located in the positive region of PC1, crude fat, crude protein, and crude ash in the negative region of PC1, indicated that they make a relatively significant contribution.

The substrate mixtures generally resulted in improved performances of BFS larvae growth, compared with the individual substrates, M0, and M100. Pure SCR has previously been shown to be a poor substrate for larval development, which may be due to its high fiber and high crude protein content [42]. BSFL could grow on by-products characterized by a high fiber diets; however, it would obtain a low growth rate, larval mortality, and final larval mass [42,49]. Larvae grown on rice straw had prepupal masses that were 84%, 71%, 79%, and 77% lower than larvae grown on KW, banana peels, brewer’s waste, and fecal sludge, respectively [50,51]. The fiber content of the SCR (Table 2) was similar to other values [1,7]. In addition, the degradation of the fiber by BSFL previously recorded in rice straw [23] and corncobs [52] was similar to the results found in the present study.

Although SCR is inherently difficult to digest by BSFL, on the positive side, it can improve loosening when mixed with KW. The presence of KW partly enhanced the protein of larval feed substrates and reduced the fiber components with low palatability. Lim et al. [15] proposed the co-digestion of waste coconut endosperm and SCR by BSFL, which confirmed the feasibility of establishing a positive synergistic effect for the biological growth of BSFL. Palma et al. [17] found that larvae harvest mass and yield increased with decreasing C/N ratio, and this can be managed to enhance bioconversion of lignocellulosic food waste. BSFL were fed four rates of dairy manure, and it was found that those fed less weighed less than those fed more [48]. The addition of rice bran can promote the transformation of chicken manure and pig manure by BSFL [22]. *Tenebrio molitor* larvae were fed on feed of 5–10% crude fiber; larvae in later instars reached optimal levels of growth, development and respiration [18]. It should be stated that each formulation included about 30% SCR and 70% KW, and these were the waste values that supported higher BSFL performance.

For the nutritional imbalance caused by a single substance, mixing multiple waste types (co-conversion) can solve this problem and produce a more nutritious and balanced feed for BSFL growth. Mixing manure and fecal sludge with food waste and other organic substrates (e.g., dairy manure and SCR) increased the larval mass to a greater extent than single waste types [7,24]. The protein conversion ratio for the mixture of abattoir waste and fruit and vegetables was higher than that of the pure substrates [27].

At present, much organic waste has not been fully developed, utilized, and recycled, causing it to eventually become environmental pollutants. The formulation of applicable waste mixtures based on nutrients requires the actual determination of composition using elements that are related to BSFL growth. Co-conversion by BSFL can add value to the organic waste, by converting it into insect products potentially suitable for varied applications, for example in animal feed and biodiesel.

## 5. Conclusions

A co-conversion technique of SCR and KW was established, which provided a practical and promising method for converting fibrous organic waste into the biomass of BSFL. The highest larval biomass (30.32 g fresh and 11.38 g dry mass), bioconversion rate (18.54%), larval crude lipid (45.91%), and lower FCR (2.51) were obtained when BSF larvae were fed with M30 (30% SCR and 70% KW). Therefore, co-conversion is a promising technique for the utilization of cellulose-rich organic matter in larval feed to reduce the impact of environmental organic waste. Further study will explore the nutrient requirements of BSFL, which correlates them with their growth performance.

## Figures and Tables

**Figure 1 insects-13-00023-f001:**
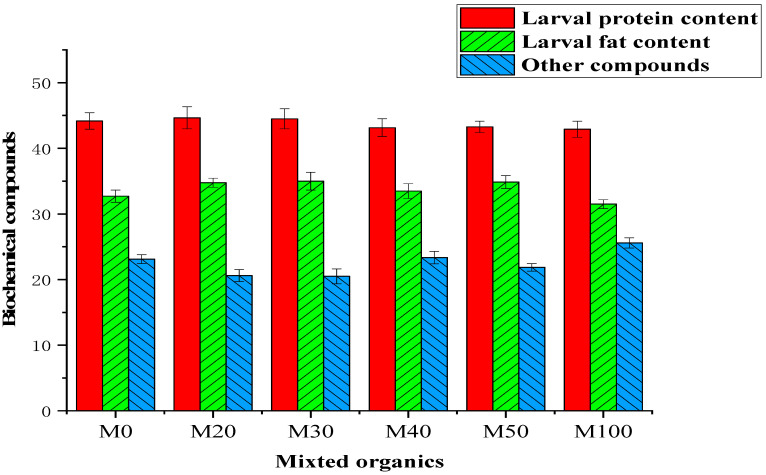
Nutritional profiles (%) of BSFL biomasses reared using various mixed organics (*n* = 3).

**Figure 2 insects-13-00023-f002:**
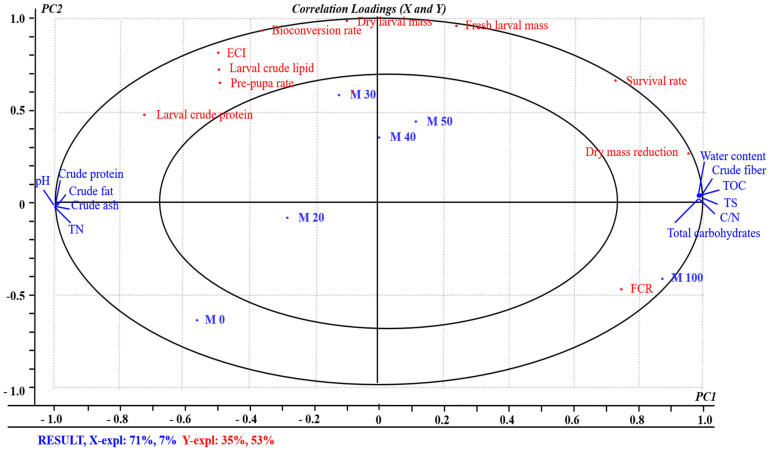
The correlation between the nutritional constituents of mixed organics and growth performance of BSFL from the PLSR correlation loading plot for samples. Six feed mixtures of SCR:KW were formulated: (0:100) M0, (20:80) M20, (30:70) M30, (40:60) M40, (50:50) M50, (100:0) M100. TOC: total organic carbon; TN: total nitrogen; TS: total sulfur; C/N ratio: carbon/nitrogen; FCR: feed conversion ratio; ECI: conversion efficiency of ingested food.

**Table 1 insects-13-00023-t001:** Comparison of selected parameters of black soldier fly conversion experiments.

References	Feed Source	Optimal Ratio	Survival Rate (%)	Fresh Larval Mass (g)	Dry Larval Mass (g/%)	Dry Mass Reduction (%)	Bioconversion Rate (%)	Temperature (°C)	Humidity (%)
Present study	SCR and KW	3:7	81.50–99.50	0.10–0.15	0.04–0.06 g	32.71–58.36	13.04–18.54	28–30	70
Rehman, Rehman, Cai, Zheng, Xiao, Somroo, Wang, Li, Yu and Zhang [7]	Dairy manure and SCR	2:3	89.50–98.40	0.06–0.10	21.4–26.5%	26–72	6.3–15.2	27	60–70
Zheng, Hou, Li, Yang, Li and Yu [23]	Restaurant waste and rice straw	7:3	NA	NA	NA	NA	NA	27	70
Isibika, et al. [25]	Fruit peels with fish waste	3:1	66.0–99.7	0.14–0.18	NA	NA	9.4–13.8	NA	80
Rehman, Cai, Xiao, Zheng, Wang, Soomro, Zhou, Li, Yu and Zhang [21]	Dairy manure and chicken manure	4:6	89.45–98.35	0.05–0.10	10.29–22.56	43.17–55.04	4.19–9.88	27	60–70
Lim, Mohd-Noor, Wong, Lam, Goh, Beniers, Oh, Jumbri and Ghani [15]	Waste coconut endosperm and SCR	3:2	NA	NA	NA	NA	NA	28–30	65–70
Nyakeri, Ayieko, Amimo, Salum, Ogola and Feed [24]	Fecal sludge with organic waste	7:3	NA	NA	NA	NA	3.07–4.67	28	65
Lopes, et al. [26]	Bread waste and aquaculture waste	<15:85	65.4–88.5	0.12–0.16	NA	41.7–46.3	14.9–18.1	28	45
Lalander, et al. [27]	Abattoir waste and fruits & veg	1:1	96.3	NA	NA	14.5	14.2	28	NA
Pliantiangtam, et al. [28]	Coconut endosperm and SCR	5:5	NA	0.10	NA	NA	NA	28	NA

(NA: not available).

**Table 2 insects-13-00023-t002:** Nutritional constituents of mixed organics between SCR and KW.

Feeding Mixture	Water Content (%)	pH	Crude Protein (%)	Crude Fat (%)	Crude Fiber (%)	Total Carbohydrates (%)	Crude Ash (%)	TOC (%)	TN (%)	TS (%)	C/N Ratio
M0	78.63 ± 0.02	6.30 ± 0.21	25.41 ± 1.81	13.37 ± 0.74	12.34 ± 1.03	28.92 ± 1.12	10.52 ± 0.34	39.79 ± 1.16	4.41 ± 0.10	5.47 ± 0.09	9.02 ± 0.06
M20	78.87 ± 0.02	6.20 ± 0.20	24.97 ± 1.63	12.53 ± 0.66	14.30 ± 0.99	29.16 ± 1.14	9.28 ± 0.31	41.02 ± 1.13	4.25 ± 0.12	5.52 ± 0.09	9.75 ± 0.11
M30	78.99 ± 0.03	6.15 ± 0.18	24.75 ± 1.54	12.11 ± 0.60	15.29 ± 0.96	29.28 ± 1.15	8.66 ± 0.27	41.63 ± 1.10	4.17 ± 0.13	5.55 ± 0.09	10.12 ± 0.14
M40	79.11 ± 0.03	6.10 ± 0.15	24.54 ± 1.38	11.69 ± 0.58	16.27 ± 0.95	29.41 ± 1.16	8.04 ± 0.25	42.25 ± 1.08	4.09 ± 0.14	5.58 ± 0.09	10.49 ± 0.18
M50	79.23 ± 0.03	6.05 ± 0.13	24.32 ± 1.10	11.28 ± 0.54	17.26 ± 0.94	29.53 ± 1.18	7.42 ± 0.23	42.86 ± 1.06	4.01 ± 0.15	5.61 ± 0.10	10.86 ± 0.22
M100	79.84 ± 0.05	5.80 ± 0.11	23.24 ± 0.34	9.19 ± 0.43	22.18 ± 0.89	30.15 ± 1.22	4.33 ± 0.16	45.94 ± 0.96	3.62 ± 0.17	5.75 ± 0.11	12.70 ± 0.32

Six feed mixtures of SCR:KW were formulated: (0:100) M0, (20:80) M20, (30:70) M30, (40:60) M40, (50:50) M50, (100:0) M100. TOC: total organic carbon; TN: total nitrogen; TS: total sulfur; C/N ratio: carbon/nitrogen; values are in mean ± S.E; *n* = 3 (30.32 and 11.38), secondly, the value increased rapidly from M40 (30.48 and 11.33) to M100 (25.38 and 8.03). The highest dry mass of BSFL was observed in M30 and M40.

**Table 3 insects-13-00023-t003:** Survival rate, prepupal rate, and larval production of BSFL fed on SCR and KW and their co-digestion mixtures. Dry mass reduction, bioconversion, and FCR of BSF converting the mixtures.

Feeding Mixture	Water Content (%)	pH	Crude Protein (%)	Crude Fat (%)	Crude Fiber (%)	Total Carbohydrates (%)	Crude Ash (%)	TOC (%)
M0	78.63 ± 0.02	6.30 ± 0.21	25.41 ± 1.81	13.37 ± 0.74	12.34 ± 1.03	28.92 ± 1.12	10.52 ± 0.34	39.79 ± 1.16
M20	78.87 ± 0.02	6.20 ± 0.20	24.97 ± 1.63	12.53 ± 0.66	14.30 ± 0.99	29.16 ± 1.14	9.28 ± 0.31	41.02 ± 1.13
M30	78.99 ± 0.03	6.15 ± 0.18	24.75 ± 1.54	12.11 ± 0.60	15.29 ± 0.96	29.28 ± 1.15	8.66 ± 0.27	41.63 ± 1.10
M40	79.11 ± 0.03	6.10 ± 0.15	24.54 ± 1.38	11.69 ± 0.58	16.27 ± 0.95	29.41 ± 1.16	8.04 ± 0.25	42.25 ± 1.08
M50	79.23 ± 0.03	6.05 ± 0.13	24.32 ± 1.10	11.28 ± 0.54	17.26 ± 0.94	29.53 ± 1.18	7.42 ± 0.23	42.86 ± 1.06
M100	79.84 ± 0.05	5.80 ± 0.11	23.24 ± 0.34	9.19 ± 0.43	22.18 ± 0.89	30.15 ± 1.22	4.33 ± 0.16	45.94 ± 0.96

Six feed mixtures of SCR:KW were formulated: (0:100) M0, (20:80) M20, (30:70) M30, (40:60) M40, (50:50) M50, (100:0) M100. (Average ± SE; *n* = 3). Average values followed by the same letters within a column do not differ significantly (*p* < 0.05). FCR: feed conversion ratio; ECI: conversion efficiency of ingested food.

## Data Availability

All relevant data are within the paper and its Appendix A.

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
