# Peer review of "Conversion of Mixtures of Soybean Curd Residue and Kitchen Waste by Black Soldier Fly Larvae (Hermetia illucens L.)"

_insects, 2021, doi:10.3390/insects13010023_

Round 1

Reviewer 1 Report

The authors have addressed all the of the points I made in my previous review, and have made extensive edits to the original submission. I would like to thank the authors for their hard work!

Author Response

Thanks very much for your positive comments.

Reviewer 2 Report

The manuscript was greatly improved from its earliest version. The authors took in account the majority of the queries from the Reviewers, adding important details in all the manuscript and carrying out a strong revision of the English style that now is more fluent and readable.

Few concerns remain:

  1. In figure 1 still no statistics are reported, table S1 is not available, as claimed from the authors in the rebuttal letter, and was not visualized. I suggest the authors to make graph with histograms reporting SE bars.
  2. The number of replicates (N=3) is barely acceptable

References need a careful check (journal is almost always missing) and a revision of the format in some case.

Line 340-344, this sentence should be deleted or changed, this was pointed out before. What does emerge from the analysis in brief?

Line 380-386, I disagree with the comments of the authors about the changes in chemical composition of the larvae determined by the ration of the substrates used: as possible to see in Fig 1, the chemical composition doesn’t change so much regardless of the diet used.

Line 426, change in “Lalander and coworkers…”

Line 438 “The waste mass reduction of the present study on dry mass bases was 32.71-58.36%” unclear sentence, give more detail

Line 450, avoid citation of figures in Discussion section. This should be restricted to Results section.

Author Response

The manuscript was greatly improved from its earliest version. The authors took in account the majority of the queries from the Reviewers, adding important details in all the manuscript and carrying out a strong revision of the English style that now is more fluent and readable.

Reply: Thanks very much for your positive comments.

Few concerns remain:

  1. In figure 1 still no statistics are reported, table S1 is not available, as claimed from the authors in the rebuttal letter, and was not visualized. I suggest the authors to make graph with histograms reporting SE bars.

Reply: Thanks very much for the professional suggestion again, in the revision we have made graph with histograms reporting SE bars. (See page 9, line 236, figure 1) And we have also modified Table S1 as you suggested. (See page 17, line 572, Table S1)

  1. The number of replicates (N=3) is barely acceptable

Reply: Thanks very much for your kind reminder. In order to maintain uniformity of data, so we used the number of replicates (N=3), and the surveys were performed in triplicate for each pure feed and mixture feed. (See page 5, line 129-130)

  1. References need a careful check (journal is almost always missing) and a revision of the format in some case.

Reply: Thanks for your kind reminder. We have checked the entire references and corrected the format in the revision. (See page 14-16, line 439-561)

  1. Line 340-344, this sentence should be deleted or changed, this was pointed out before. What does emerge from the analysis in brief?

Reply: Thanks very much for the valuable suggestion, in the revision we have deleted the sections as you suggested. (See page 10, line 264)

  1. Line 380-386, I disagree with the comments of the authors about the changes in chemical composition of the larvae determined by the ration of the substrates used: as possible to see in Fig 1, the chemical composition doesn’t change so much regardless of the diet used.

Reply: Thanks very much for your professional comments. As shown in Figure 2, the mixed organics on M30 on the negative region of PC1, have good correspondence with the Larval crude protein and Larval lipid. So, we conclude the chemical composition of the larvae determined by the ration of the substrates used. (See page 11, line 293-300)

  1. Line 426, change in “Lalander and coworkers…”

Reply: Thanks for your kind suggestion. In the revision, we have modifies the specification of the refer to reference according to the suggestion. (See page 11, line 331)

  1. Line 438 “The waste mass reduction of the present study on dry mass bases was 32.71-58.36%” unclear sentence, give more detail

Reply: Thanks very much for your comments, in the revision we have modified the the sentence as you suggested. (See page 11, line 341)

  1. Line 450, avoid citation of figures in Discussion section. This should be restricted to Results section.

Reply: Thanks for your kind reminder. In the revision, we have deleted the citation of figures in discussion section according to the suggestion. (See page 10-13, 4 Discussion, line 260)

This manuscript is a resubmission of an earlier submission. The following is a list of the peer review reports and author responses from that submission.

Round 1

Reviewer 1 Report

The research presented in the manuscript concerns the rearing of black soldier fly larvae upon substrates comprising mixtures of soybean curd residue (SCR) and kitchen waste (KW). The effect of altering the substrate composition upon the characteristics of the black soldier fly larvae is explored. The results show that the best combination of SCR and FW is a 30:70 ratio by weight. Further discussion is given to the correlation of the BSFL larvae composition to the different substrate properties.

The research is within an area of great interest and the approach to presenting the data is interesting. It would make a good contribution to the body of work. However, the language used within the manuscript needs considerable improvement. I have great respect for authors who are not writing in their first language, but there are multiple points in the manuscript where the meaning of the words is ambiguous to the point of being confusing. In addition to the points raised below, I strongly advise that the authors proofread the manuscript thoroughly.

  • The fist point is a major one: the frequent and interchangeable use of “BSFL”, “BASL”, “BSF larvae”, “BAS larvae”, “BAS larval”, and in the conclusion, “Hermetia illucens larvae”. Please be consistent throughout. The only time that BAS is defined is after “BSF larvae” in the Simple Summary. Therefore, I assume BAS is the same as BSF? It makes the manuscript feel as though it was written by 4 different people and checked for consistency upon compiling.
  • Likewise, weight and mass are interchanged. Most notably this occurs in the equations. Weight and mass are, technically speaking, two different physical properties. Please check for consistency on this.
  • Another major concern I had was the focus on referenced literature within the discussion. I appreciate that the authors are comparing the present study to other studies, but the writing style makes it hard to distinguish what is their own work, and what is others work. It would benefit the reader greatly to better and more clearly emphasise the differences. Please consider re-writing this section to give much greater emphasis on the work from this study.
  • Furthermore, there appears to be duplication of discussion points between the two sub-sections within the discussion section. Please amend the text to highlight the different discussion points within the sub-sections or combine into one discussion.
  • Equations 4 and 6 appear to be rather similar. Can the difference be explained in more detail, please?
  • Equation 5 appears to be the inverse of either 4, or 6. Is there another, more significant difference in the equations?
  • The term conversion appears to have one or two meanings within the manuscript. In the simple summary it is used as though the act of mixing the SCR and FW is conversion. Whereas later on, it appears to be used more in terms of the act of the BSFL consuming the substrates. Please check and clarify.
  • Table 3: The error bars for survival rate indicate that survival rates in excess of 100% were achieved. I assume this is a quirk of the statistics, but please explain it in the manuscript.
  • Line 47: the statement regarding FW nutritional content needs a reference.
  • Line 50: please specify the breeding industry that the SCR and FW are applied in.
  • Lines 73-75: I could not understand this sentence, please review.
  • Lines 92-94: I could not understand this sentence, please review.
  • In section 2.1, the number of BSFL is referred to as 1200, but in section 2.2, it is 200. Please clarify which number is correct, or why they are different.
  • Line 137: does “when feeding” mean that substrate was added throughout the 12 days, or that the substrate was only added at the start, and the BSFL left to feed for 12 days? Please clarify.
  • Line 159: a range is referred to in this sentence, but only one value given. What is the other end of the range?
  • Line 204: “decrease” is used in this sentence, but should it be “increase”?
  • Lines 213-215: Should all the comparisons be reversed? For example, “highest” changed to “lowest”, and vice versa. Likewise, should “decreased” be changed to “increased”? If not, please review the definition of ECI as it may need a clearer explanation.
  • Line 215: Should “29.07%” be “19.07%”?
  • Line 219: The value “13.40%”, where did this come from?
  • Line 223: please give the value for conversion rate of the dairy manure in the reference.
  • In section 3.4 please consider modifying the language regarding the changes. For example, “drastically advanced” is used to describe a change from 30.51% to 32.71%, or to 35%. I would not necessarily term that drastic.
  • Lines 228 & 229: there are % symbols missing after 44.17 and 44.65.
  • Line 276-278: I could not understand this sentence, please review.
  • Line 307 and 319: Is the referencing style consistent? These are an example of a reference written with 4 names in the text, and another written “et al” after the first name. Unless it is journal policy, please consider revising so that papers with more than 2 authors are referenced in text as “*name* et al”.
  • Lines 312-314: I could not understand this sentence, please review.
  • Lines 315-316: I could not understand this sentence, please review.
  • Line 315: What is “bioconversion rate dry”? I think I know, but it is the first time I was aware of this particular phrase being used, which indicates that different wording was used elsewhere.
  • Lines 355-364: Throughout these lines, crude appears to be spelled “curde”.
  • Line 373: what does “improved buffering” mean? Please explain within the text.
  • In the conclusion, the hazardous wastes are referred to for the first time. What hazardous wastes?
  • Lastly, I believe there is a typo in the first affiliation of the authors: “College of Food Scinence…”.

Author Response

R1- Replies to comments on insects-1436864

The research presented in the manuscript concerns the rearing of black soldier fly larvae upon substrates comprising mixtures of soybean curd residue (SCR) and kitchen waste (KW). The effect of altering the substrate composition upon the characteristics of the black soldier fly larvae is explored. The results show that the best combination of SCR and FW is a 30:70 ratio by weight. Further discussion is given to the correlation of the BSFL larvae composition to the different substrate properties.

The research is within an area of great interest and the approach to presenting the data is interesting. It would make a good contribution to the body of work. However, the language used within the manuscript needs considerable improvement. I have great respect for authors who are not writing in their first language, but there are multiple points in the manuscript where the meaning of the words is ambiguous to the point of being confusing. In addition to the points raised below, I strongly advise that the authors proofread the manuscript thoroughly.

Reply: Thank you for your comments. We carefully checked the full text as you suggested. And, the English of this manuscript has been revised by a native English-speaker engaged through the auspices of a professional proofreading service. Language Revision Certificate is shown in second page of this response letter.

  1. The fist point is a major one: the frequent and interchangeable use of “BSFL”, “BASL”, “BSF larvae”, “BAS larvae”, “BAS larval”, and in the conclusion, “Hermetia illucens larvae”. Please be consistent throughout. The only time that BAS is defined is after “BSF larvae” in the Simple Summary. Therefore, I assume BAS is the same as BSF? It makes the manuscript feel as though it was written by 4 different people and checked for consistency upon compiling.

Reply: Thanks for your kind reminder. So sorry about the original expression “BASL” and “BAS larval” are misleading, however, “BAS larvae” and “BSFL” are correct. We have checked the entire test and corrected in the revision.

  1. Likewise, weight and mass are interchanged. Most notably this occurs in the equations. Weight and mass are, technically speaking, two different physical properties. Please check for consistency on this.

Reply: Thanks very much for your professional suggestion. In order to maintain consistency, we have replaced “weight” by “mass” thoughout the text in the revision. (See page 3, line 67)

  1. Another major concern I had was the focus on referenced literature within the discussion. I appreciate that the authors are comparing the present study to other studies, but the writing style makes it hard to distinguish what is their own work, and what is others work. It would benefit the reader greatly to better and more clearly emphasise the differences. Please consider re-writing this section to give much greater emphasis on the work from this study.

Reply: Thanks very much for your professional suggestion. The English of this manuscript has been revised by a native English-speaker engaged through the auspices of a professional proofreading service. The paper and the discussion should make our work accessible to the scientific community.

  1. Furthermore, there appears to be duplication of discussion points between the two sub-sections within the discussion section. Please amend the text to highlight the different discussion points within the sub-sections or combine into one discussion.

Reply: Thanks very much for the valuable suggestion, in the revision we have highlighted the subheading of sub-sections as you suggested. (See page 12, line 354)

  1. Equations 4 and 6 appear to be rather similar. Can the difference be explained in more detail, please?

Reply: Thanks for your kind suggestion. In the revision, we have modifies the equations 4 and 6 according to the suggestion. The difference are the denominator of feed added and mass of ingested feed (See page 6, line 167-169)

  1. Equation 5 appears to be the inverse of either 4, or 6. Is there another, more significant difference in the equations?

Reply: Thanks for your kind suggestion. In the revision, we have modifies the equations 5 according to the suggestion. (See page 6, line 168)

  1. The term conversion appears to have one or two meanings within the manuscript. In the simple summary it is used as though the act of mixing the SCR and FW is conversion. Whereas later on, it appears to be used more in terms of the act of the BSFL consuming the substrates. Please check and clarify.

Reply: Thanks very much for your comments, in the revision we have modified the the sentence as you suggested. (See page 2, line 13-14)

  1. Table 3: The error bars for survival rate indicate that survival rates in excess of 100% were achieved. I assume this is a quirk of the statistics, but please explain it in the manuscript.

Reply: Thanks very much for your comments, in the revision we have discussed the survival rates in 4.1. Process performance. (See page 10, line 283-295)

  1. Line 47: the statement regarding FW nutritional content needs a reference.

Reply: Thanks for your kind reminder. In the revision, we have added the reference  according to the suggestion. (See page 2, line 46)

  1. Line 50: please specify the breeding industry that the SCR and FW are applied in.

Reply: Thanks for your kind reminder. In the revision, we have added the explanation “e.g. livestock and poultry breeding” according to the suggestion. (See page 2, line 47-48)

  1. Lines 73-75: I could not understand this sentence, please review.

Reply: Thanks very much for the valuable comment, in the revision we have rephrase the sentence as you suggested. (See page 3, line 68-71)

  1. Lines 92-94: I could not understand this sentence, please review.

Reply: Thanks very much for the valuable comment, in the revision we have rephrase the sentence as you suggested. (See page 3, line 90-92)

  1. In section 2.1, the number of BSFL is referred to as 1200, but in section 2.2, it is 200. Please clarify which number is correct, or why they are different.

Reply: Thanks for your kind reminder. In the revision, we have added the explanation “sixth group, total 1200” according to the suggestion. (See page 5, line 126)

  1. Line 137: does “when feeding” mean that substrate was added throughout the 12 days, or that the substrate was only added at the start, and the BSFL left to feed for 12 days? Please clarify.

Reply: Thanks very much for the valuable suggestion, in the revision we have rephrase the sentence as you suggested. (See page 5, line 131-132)

  1. Line 159: a range is referred to in this sentence, but only one value given. What is the other end of the range?

Reply: Thanks for your kind reminder. The original expression was incomplete; In the revision, we have added the range according to the suggestion. (See page 5, line 154)

  1. Line 204: “decrease” is used in this sentence, but should it be “increase”?

Reply: Thanks for your kind reminder. The original expression was misleading; In the revision, we have replaced “decreased” by “increased” according to the table 3. (See page 8, line 203)

  1. Lines 213-215: Should all the comparisons be reversed? For example, “highest” changed to “lowest”, and vice versa. Likewise, should “decreased” be changed to “increased”? If not, please review the definition of ECI as it may need a clearer explanation.

Reply: Thanks for your kind reminder. Efficiency of conversion of ingested food (ECI) is calculated as change in body weight divided by feeding amount. Therefore, may be the original expression was inerrable. (See page 8, line 210-211)

  1. Line 215: Should “29.07%” be “19.07%”?

Reply: Thanks for your kind reminder. The original expression was misleading; In the revision, we have replaced “29.07%” by “19.07%” according to the table 3. (See page 8, line 211)

  1. Line 219: The value “13.40%”, where did this come from?

Reply: Thanks for your kind reminder. The original expression was misleading; In the revision, we have replaced “13.40%” by “15.14%” according to the table 3. (See page 8, line 215)

  1. Line 223: please give the value for conversion rate of the dairy manure in the reference.

Reply: Thanks for your kind reminder. We have added the “14.60%” in the revision. (See page 3, line 218)

  1. In section 3.4 please consider modifying the language regarding the changes. For example, “drastically advanced” is used to describe a change from 30.51% to 32.71%, or to 35%. I would not necessarily term that drastic.

Reply: Thanks for your professional and kind reminder. The original expression was misleading; we have replaced “drastically advanced” by “significant change” in the revision. (See page 8, line 231)

  1. Lines 228 & 229: there are % symbols missing after 44.17 and 44.65.

Reply: Thanks for your reminder. We have added the “%” in the revision. (See page 8, line 229)

  1. Line 276-278: I could not understand this sentence, please review.

Reply: Thanks very much for the valuable suggestion, in the revision we have rephrase the sentence as you suggested. (See page 10, line 280-282)

  1. Line 307 and 319: Is the referencing style consistent? These are an example of a reference written with 4 names in the text, and another written “et al” after the first name. Unless it is journal policy, please consider revising so that papers with more than 2 authors are referenced in text as “*name* et al”.

Reply: Thanks for your professional and kind reminder. The original reference was inconsistent; in the revision we have corrected the reference ““*name* et al”” as you suggested throughout the text. (See page 11, line 306)

  1. Lines 312-314: I could not understand this sentence, please review.

Reply: Thanks very much for the valuable suggestion, in the revision we have rephrase the sentence as you suggested. (See page 11, line 312-314)

  1. Lines 315-316: I could not understand this sentence, please review.

Reply: Thanks very much for the valuable suggestion, in the revision we have rephrase the sentence as you suggested: The pre-pupal rate of BASL fed on pure substrate (SCR 80.36% and KW 83.48%, respectively) was significantly lower than that in mixed substrate group, which may be related to the advantage of the mixture. (See page 11, line 316-319)

  1. Line 315: What is “bioconversion rate dry”? I think I know, but it is the first time I was aware of this particular phrase being used, which indicates that different wording was used elsewhere.

Reply: Thanks for your professional and kind reminder. The original expression was misleading; we have deleated the word “dry” in the in the revision. (See page 11, line 315)

  1. Lines 355-364 (in the original manuscript): Throughout these lines, crude appears to be spelled “curde”.

Reply: Thanks for your professional and kind reminder. The original expression was misleading. In the revision, we have replaced “curde” by “crude” throughout the text. (See page 12, line 358)

  1. Line 373 (in the original manuscript): what does “improved buffering” mean? Please explain within the text.

Reply: Thanks for your professional and kind reminder. The original expression was misleading; we have replaced “buffering” by “loosening”. (See page 12, line 375)

  1. In the conclusion, the hazardous wastes are referred to for the first time. What hazardous wastes?

Reply: Thanks for your professional and kind reminder. The original expression was misleading; we have replaced “hazardous wastes” by “organic wastes”. (See page 13, line 406)

  1. Lastly, I believe there is a typo in the first affiliation of the authors: “College of Food Scinence…”.

Reply: Thanks for your kind reminder. The original expression was misleading; we have replaced “Scinence” by “Science”. (See page 1, line 5)

Reviewer 2 Report

You are in the good way. Please include the clarifications and the job willl be fine.

I will propose a major revision, but it looks like very good.

Author Response

R1- Replies to comments on insects-1436864

Title suggestion: Conversion of mixture of soybean curd residue and kitchen waste by black soldier fly larvae (Hermetia illucens L.)

Reply: Thanks very much for the kind suggestion, in the revision we have corrected the title as you suggested. (See page 1, line 2-3)

  1. CHECK BSFL and BAS Larvae, to decide to use one of the two.

Reply: Thanks for your kind reminder. So sorry about the original expression “BASL” and “BAS larval” are misleading, however, “BAS larvae” and “BSFL” are correct. We have checked the entire test and corrected in the revision.

  1. Line 11: to change with “is a viable solution for food waste management and can provide…”

Reply: Thanks very much for the valuable suggestion, in the revision we have corrected the sentence as you suggested. (See page 2, line 12-13)

  1. Line 14: To delete “High percentage of fiber limits the performance of the larvae. This sentence does not make sense considering the following sentences”.

Reply: Thanks very much for the valuable suggestion, in the revision we have deleted the sentence as you suggested. (See page 2, line 14)

  1. Line 15: to change “For the nutritional…. this problem and provide “, with “Not-balanced diet caused by utilization of single substrate could be solved using a mix of different waste and formulating a more balanced diet and provide…...”

Reply: Thanks very much for the professional suggestion, in the revision we have rephrase the sentence as you suggested. (See page 2, line 13-15)

  1. Line 19: “Main results of this study are:…”

Reply: Thanks very much for the professional suggestion, in the revision we have added the sentence as you suggested. (See page 2, line 13-15)

  1. Line 21 to 23: to delete all this sentence from “Therefore, the conversion…. ardous waste”. This sentence is does not make sense with the main results.

Reply: Thanks very much for the professional suggestion, in the revision we have deleted the sentence as you suggested. (See page 2, line 20)

Abstract

  1. Line 24: soybean curd is not a high fiber cellulose rich organic waste. A high fiber substrate is around 30-40% of fiber. Your data presented only 12-15% of fiber.

Reply: Thank very much for your comment. Yes, soybean curd is not a high fiber cellulose rich organic waste. The soybean curd residue (SCR) should be a cellulose-rich organic waste (about 22% fiber in the presented study). (Table 2)

  1. It is better to delete “from cellulose-rich”

Reply: Thanks very much for the professional suggestion, in the revision we have deleted the “cellulose-rich” as you suggested. (See page 2, line 22)

  1. Line 68: To delete: ”However,…. of the larvae”.

Reply: Thanks very much for the professional suggestion, in the revision we have deleted the sentence as you suggested. (See page 3, line 66)

  1. Line 69: to include this sentence: “ The pH seems play an important role on the BAS larvae and the larvae can modulate it to maintain the best condition (You can cite

Meneguz, M., Gasco, L., & Tomberlin, J. K. (2018). Impact of pH and feeding system on black soldier fly (Hermetia illucens, L; Diptera: Stratiomyidae) larval development. PloS one, 13(8), e0202591. And Ma, J., Lei, Y., Rehman, K. U., Yu, Z., Zhang, J., Li, W., ... & Zheng, L. (2018). Dynamic effects of initial pH of substrate on biological growth and metamorphosis of black soldier fly (Diptera: Stratiomyidae). Environmental entomology, 47(1), 159-165.

Reply: Thanks very much for the professional suggestion, in the revision we have deleted the sentence and the references as you suggested. (See page 3, line 70-72)

  1. Table 2: I suggest adding Ash content if it possible in the table’s values

It is very determinant in the final concentration of heavy metal and other ash on the larval composition.

Reply: Thanks for your kind reminder. The “Crude ash (%)” was shown in Figure 2 in the revision (See page 5, line 199)

  1. I don’t understand why you put Table 3 with result in this part near to table 2.

I will move this table at the end of the paper or in the Results part.

Reply: Thanks very much for the professional suggestion, in the revision we have moved Table 3 to the results part as you suggested. (See page 8, line 219)

Materials and Methods

  1. To insert title with “Experimental model”: replicates, larvae per crate and climatic condition and all the information about the experiment in this paragraph

I think you perform different test, it could be better to divide in paragraph them.

Reply: Thanks very much for the professional suggestion, in the revision we have rephrased the related sentence as you suggested. (See page 5, line 124-129)

  1. Line 173: use the same terminology of the Table 3 to simplify the understanding of the reader; for example use instead of “weight of BSF larval” with “Dry larval weight” Include in the table 3 the Dry Feed intake to understand and check the result of FCR.

Reply: Thanks for your kind reminder. We have replaced “weight of BSF larval” by “Dry larval weight” and checked the result of FCR. (See page 8, line 219)

  1. Line 116 to 127:

Move all the part about experimental model to a second paragraph and let in “Raw material” only chemical characterization, origin, and other information as you reported in the paragraph.

Reply: Thanks very much for the professional suggestion, in the revision we have rephrased the related sentence as you suggested. (See page 5, line 124-130)

  1. Line 161: change prepupa rate with prepupal rate

Reply: Thanks for your kind reminder. We have replaced “prepupa rate” by “prepupal rate” throughout the text. (See page 7, line 163)

  1. Move table 2 in the result area as table 3.

Reply: Thanks very much for the professional suggestion, in the revision we have moved table 2 to the results part as you suggested. (See page 7, line 199)

  1. Line 224 to 234: To include a table with the values of Crude Protein, Crude Fat, Ash , Fiber of BSF larvae.

Reply: Thanks very much for the professional suggestion, in the revision we have added Table S1 as you suggested. However, only crude protein and crude fat of BSFL was shown. Sincerely thank you for your helpful comments, and we have consider this for the next experiments.(See page 8, line 228; See page 17, line 573;)

Discussion

  1. I will do a revision on the discussion after the request reported in this summary

Thanks

Reply: Thank you for your valuable comments.

Reviewer 3 Report

The manuscript deals with an important topic as the use of waste substrates used in different mixtures for rearing H. illucens in order to evaluate the effects on the growth performances. Regrettably, the paper has a limited scientific values (too small number of replicates) and is poorly written (several sentence are obscure).

Main critical points in detail:

  1. The methodology and results suffer for the limited number of replicates carried out (3 per treatment).
  2. Several oversights and misspelled words are present across the manuscript. Several section are cumbersome: a deep review from a moth-tongue entomologist is a “must”.
  3. The background of the study is not well introduced, introduction is fragmentary.
  4. In introduction, the concept of “kitchen waste” should be better introduced as is something difficult to standardize: kitchen waste from China are different from the ones in USA or Europe for example.
  5. Statistics are missing important details.
  6. In the abstract several acronyms are not defined or could be avoided.
  7. Figure 1 show similar percentage of the components regardless the diet used. No standard error is reported nor letters or asterisk that could indicate a significant different between treatments.
  8. Discussion need a strong rewriting as is cumbersome and could be implemented with few references that addressed a very strictly-related topic, see for example:

Mahmood, S., Zurbrügg, C., Tabinda, A. B., Ali, A., & Ashraf, A. (2021). Sustainable Waste Management at Household Level with Black Soldier Fly Larvae (Hermetia illucens). Sustainability13(17), 9722.

El-Hack, A., Mohamed, E., Shafi, M. E., Alghamdi, W. Y., Abdelnour, S. A., Shehata, A. M., ... & Ragni, M. (2020). Black soldier fly (Hermetia illucens) Meal as a promising feed ingredient for poultry: A comprehensive review. Agriculture10(8), 339.

  1. Finally, overall is unclear which are the main findings obtained in this study, the author should highlight them better.

Other issues

Line 14 error in the acronym, please check

Line 21, delete “M30” as not clarified before, and avoid the parenthesis

Line 29, delete “PLSR” as not present in the abstract elsewhere

Line 29-32, revise this part as unclear.

Line 41-41, change “regarded” with “treated”

Line 43-44, when?

Line 50, missing space

Lines 66-68, explain more in detail this concept.

Line 72-72 change in “Dietary fiber limits the development of larval growth and development of Tenebrio molitor (Coleoptera: Tenebrionidae) another species used to obtain protein source from wastes (reference).”

Line 76, BSFL or BASL? It’s confusing.

Lines 82-90 this part is very cumbersome and inappropriate English.

Line 112, authors should revise, it is not correct. Change in “Overall goal of this work is to develop a …”

Line 116, delete “L. (Diptera: Stratiomyidae)”

In the section “Raw materials” is missing the rearing condition of the H. illucens colony (temperatures, photoperiod, etc).

Line 128, revise misspelling.

Line 175, please be more specific, which data were analyzed by ANOVA? Delete the following sentence.

Line 260, abbreviate in H. illucens.

Line 283-285, unclear please rewrite.

Line 319-320, unclear sentence.

Line 343, delete “comparable”

Check carefully the references section as few papers have oversights (e.g. name of the species not in italic, etc)

Author Response

R1- Replies to comments on insects-1436864

The manuscript deals with an important topic as the use of waste substrates used in different mixtures for rearing H. illucens in order to evaluate the effects on the growth performances. Regrettably, the paper has a limited scientific values (too small number of replicates) and is poorly written (several sentence are obscure).

Reply: Thanks very much for your comments. In this experiment, approximately 200 of 6 days old BSFL (sixth group, total 1200) were inoculated into each vessel with the recording of a continuous date and time. Batch tests were conducted to determine the larval growth and development. The surveys were carried out in triplicate for each pure feed and mixture feed. (See page 5, line 124-130) May be the test batch and quantity are relatively normal. And, the English of this manuscript has been revised by a native English-speaker engaged through the auspices of a professional proofreading service. Language Revision Certificate is shown in second page of this response letter.

Main critical points in detail:

  1. The methodology and results suffer for the limited number of replicates carried out (3 per treatment).

Reply: Thanks very much for your comments. In this experiment, approximately 200 of 6 days old BSFL (sixth group, total 1200) were inoculated into each vessel with the recording of a continuous date and time. Batch tests were conducted to determine the larval growth and development. The surveys were carried out in triplicate for each pure feed and mixture feed. (See page 5, line 124-130)

  1. Several oversights and misspelled words are present across the manuscript. Several section are cumbersome: a deep review from a moth-tongue entomologist is a “must”.

Reply: Thanks very much for your kind suggestion. The English of this manuscript has been revised by a native English-speaker engaged through the auspices of a professional proofreading service. Language Revision Certificate is shown in second page of this response letter.

  1. The background of the study is not well introduced, introduction is fragmentary.

Reply: Thanks very much for the valuable suggestion, in the revision we have rephrase the introduction as you suggested. (See page 2, introduction)

  1. In introduction, the concept of “kitchen waste” should be better introduced as is something difficult to standardize: kitchen waste from China are different from the ones in USA or Europe for example.

Reply: Thanks very much for the valuable suggestion, in the revision we have rephrase the introduction as you suggested. (See page 2, line 42-46)

  1. Statistics are missing important details.

Reply: Thanks very much for the valuable suggestion, in the revision we have rephrased the statistical analysis as you suggested. (See page 6, line 171-175)

  1. In the abstract several acronyms are not defined or could be avoided.

Reply: Thanks very much for your remind. In the revision, we have checked and corrected the acronyms in the abstract as you suggested. (See page 2, abstract)

  1. Figure 1 show similar percentage of the components regardless the diet used. No standard error is reported nor letters or asterisk that could indicate a significant different between treatments.

Reply: Thanks very much for the professional suggestion, in the revision we have added Table S1 as you suggested. (See page 17, line 573)

  1. Discussion need a strong rewriting as is cumbersome and could be implemented with few references that addressed a very strictly-related topic, see for example:

Mahmood, S., Zurbrügg, C., Tabinda, A. B., Ali, A., & Ashraf, A. (2021). Sustainable Waste Management at Household Level with Black Soldier Fly Larvae (Hermetia illucens). Sustainability13(17), 9722.

El-Hack, A., Mohamed, E., Shafi, M. E., Alghamdi, W. Y., Abdelnour, S. A., Shehata, A. M., ... & Ragni, M. (2020). Black soldier fly (Hermetia illucens) Meal as a promising feed ingredient for poultry: A comprehensive review. Agriculture10(8), 339.

Reply: Thank you for your comment, in the revision, we have rephrased the sentence and cited the references according to the suggestion. (See page 11, line 301; line 343-344)

  1. Finally, overall is unclear which are the main findings obtained in this study, the author should highlight them better.

Reply: Thanks very much for your comments. In the revision we have rephrased and  highlight the sentence as you suggested. (See page 2, Simple Summary and Abstract)

Other issues

  1. Line 14 error in the acronym, please check

Reply: Thanks for your kind reminder. In the revision we have corrected. (See page 2, line 20)

  1. Line 21, delete “M30” as not clarified before, and avoid the parenthesis

Reply: Thanks for your kind reminder. In the revision we have deleted the “M30” and the parenthesis. (See page 2, line 20)

  1. Line 29, delete “PLSR” as not present in the abstract elsewhere

Reply: Thanks for your kind reminder. In the revision we have deleted the “PLSR”. (See page 2, line 27)

  1. Line 29-32, revise this part as unclear.

Reply: Thanks very much for your suggestion, in the revision we have rephrased the sentence as you suggested. (See page 2, line 27-29)

  1. Line 41-41, change “regarded” with “treated”

Reply: Thanks for your suggestion, In the revision, we have replaced “regarded” by “treated”. (See page 2, line 39)

  1. Line 43-44, when?

Reply: Thanks for your kind reminder. In the revision we have added the time. (See page 2, line 40)

  1. Line 50, missing space

Reply: Thanks for your kind reminder. In the revision we have corrected. (See page 2, line 48)

  1. Lines 66-68, explain more in detail this concept.

Reply: Thanks very much for your suggestion, in the revision we have revised the sentence as you suggested. (See page 3, line 70-72)

  1. Line 72-72 change in “Dietary fiber limits the development of larval growth and development of Tenebrio molitor (Coleoptera: Tenebrionidae) another species used to obtain protein source from wastes (reference).”

Reply: Thanks very much for your suggestion, in the revision we have rephrased the sentence as you suggested. (See page 3, line 68-70)

  1. Line 76, BSFL or BASL? It’s confusing.

Reply: Thanks for your kind reminder. Because of different limitations, so sorry that the original expression “BASL” and “BAS larval” are misleading, however, “BAS larvae” and “BSFL” are correct. We have checked the entire test and corrected in the revision.

  1. Lines 82-90 this part is very cumbersome and inappropriate English.

Reply: Thanks very much for your suggestion, in the revision we have rephrased the sentence as you suggested. (See page 3, line 83-93)

  1. Line 112, authors should revise, it is not correct. Change in “Overall goal of this work is to develop a …”

Reply: Thanks very much for your suggestion, in the revision we have rephrased the sentence as you suggested. (See page 3, line 98)

  1. Line 116, delete “L. (Diptera: Stratiomyidae)”

Reply: Thanks very much for your suggestion, in the revision we have deleted the “L. (Diptera: Stratiomyidae)” as you suggested. (See page 5, line 110)

  1. In the section “Raw materials” is missing the rearing condition of the H. illucens colony (temperatures, photoperiod, etc).

Reply: Thanks very much for the professional suggestion, in the revision we have added and rephrased the related information as you suggested. (See page 5, line 124-130)

  1. Line 128, revise misspelling.

Reply: Thanks a lot for your kind reminder, we have revised it in the revised manuscript. (See page 5, line 128)

  1. Line 175, please be more specific, which data were analyzed by ANOVA? Delete the following sentence.

Reply: Thanks very much for the valuable suggestion, in the revision we have rephrased the sentence as you suggested. (See page 6, line 171-173)

  1. Line 260, abbreviate in H. illucens.

Reply: Thanks a lot for your kind reminder, we have corrected it in the revised manuscript. (See page 10, line 260)

  1. Line 283-285, unclear please rewrite.

Reply: Thanks very much for the valuable suggestion, in the revision we have rephrased the sentence as you suggested. (See page 10, line 283-285)

  1. Line 319-320, unclear sentence.

Reply: Thanks very much for the valuable suggestion, in the revision we have rephrased the sentence as you suggested. (See page 11, line 321-323)

  1. Line 343, delete “comparable”

Reply: Thanks for your kind reminder. We have deleted the“comparable”as you suggested in the revision. (See page 11, line 346)

  1. Check carefully the references section as few papers have oversights (e.g. name of the species not in italic, etc)

Reply: Thanks for your kind reminder. We have carefully checked the references section and corrected as you suggested in the revision. (See page 14-16)
